# Fc Epsilon RI–Neuroimmune Interplay in Pruritus Triggered by Particulate Matter in Atopic Dermatitis Patients

**DOI:** 10.3390/ijms241411851

**Published:** 2023-07-24

**Authors:** Dina Isaifan, Sergio Crovella, Lama Soubra, Maryam Al-Nesf, Martin Steinhoff

**Affiliations:** 1Department of Biological and Environmental Sciences, College of Arts and Sciences, Qatar University, Doha P.O. Box 2713, Qatar; di1510538@qu.edu.qa (D.I.); lama.soubra@qu.edu.qa (L.S.); 2Laboratory of Animal Research Center (LARC), Qatar University, Doha P.O. Box 2713, Qatar; 3Allergy and Immunology Division, Hamad Medical Corporation, Doha P.O. Box 3050, Qatar; mariamali@hamad.qa; 4Department of Dermatology & Venereology, Weill Cornell Medicine, New York, NY 10065, USA; martin_steinhoff@web.de

**Keywords:** atopic dermatitis, air pollution, Fc-Epsilon, particulate matter, neuroimmune, pruritus

## Abstract

Atopic dermatitis (AD) is the most common chronic relapsing neuroinflammatory skin disease that is characterized by a complex and multifactorial pathophysiology. It reflects a profound interplay between genetic and environmental factors, and a recently disclosed neuroimmune dysregulation that drives skin barrier disruption, pruritus, and microbial imbalance. In terms of the key external environmental players that impact AD, air quality and itch severity linkage have been thoroughly researched. The impact of ambient air pollutants including particulate matter (PM) and AD pruritic exacerbation has been recorded despite reductions in air pollution levels in in developed countries. The developing countries have, on the contrary, experienced significant urbanization and industrialization with limited environmental protection standards in the past decades. This unprecedented construction, petrochemical industry utilization, and increment in population counts has been paired with consistent exposure to outdoor PM. This may present a key cause of AD pruritic exacerbation supported by the fact that AD prevalence has intensified globally in the past 50 years, indicating that environmental exposure may act as a trigger that could flare up itch in vulnerable persons. At the molecular level, the impact of PM on severe pruritus in AD could be interpreted by the toxic effects on the complex neuroimmune pathways that govern this disease. AD has been recently viewed as a manifestation of the disruption of both the immune and neurological systems. In light of these facts, this current review aims to introduce the basic concepts of itch sensory circuits in the neuroimmune system. In addition, it describes the impact of PM on the potential neuroimmune pathways in AD pathogenesis with a special focus on the Fc Epsilon RI pathway. Finally, the review proposes potential treatment lines that could be targeted to alleviate pruritus based on immune mediators involved in the Fc Epsilon signaling map.

## 1. Introduction

Atopic dermatitis (AD) is a chronic psycho-neuroimmune skin disease characterized by itchy eczematous lesions with debilitating impacts on patients’ quality of life. Clinical presentations of AD include erythema, lichenification, oozing, dryness, and crusting. Itch remains the mainstay feature of AD [1]. The prevalence of pruritus in AD patients ranges between 80 and 100% [2].

Atopic dermatitis has a global prevalence of 1–3% among adults, between 2–30% in children, and affects more than 280 million individuals globally [3]. This disease is manifested with chronic eczematous lesions, itching and scratching driven by evident immune changes (innate and adaptive immune system) and epidermal barrier dysfunction. AD is different in different geographical regions. In general, almost 85% of AD diagnoses occur in childhood with a substantial fraction that continues towards adulthood or relapses after remission [4]. Many reports have suggested gender equality; however, a higher prevalence in children of black ethnicity and age-sex variations have been reported in the USA. In addition, children with AD develop progressive atopy (atopic march), where AD is considered the access gate to different allergic diseases, such as food allergy, allergic rhinitis, and asthma [5].

Atopic dermatitis is a complex skin disease; multiple factors contribute to its pathogenesis where beside dysregulated adaptive and innate immunity, dysregulation of neuroimmune pathways plays a new vital role [6]. Additionally, genetic predisposition (e.g., genetic mutations in filaggrin), cell-mediated immune responses, IgE-mediated hypersensitivity, and exogenous causes, including air pollution and other environmental factors, are crucial in the pathophysiology of AD [7,8,9]. Among the major contributors to poor air quality is PM that comes in course (PM10), fine (PM2.5), and ultrafine (PM0.1) sizes. Smaller PM can easily access the skin and; therefore, drive higher risks including AD pruritus exacerbations.

Atopic dermatitis is now viewed by clinicians as an immune disease where genetic and environmental triggers interplay to activate both the immune and nervous systems. The disruption of neuroimmune signaling pathways results in distinguished symptoms of relapsing AD, including skin barrier dysfunction, chronic inflammation, and severe pruritus. Vice versa, inflammation and skin barrier dysfunction aggravate and perpetuate pruritus. Despite the many studies available on the relationship between neuroimmune signaling in AD and pruritus, few reports have explored the molecular mechanisms that govern the impact of environmental triggers on these neuroimmune pathways and their link to severe itch in AD patients. In addition, no research is available on the role of Fc Epsilon RI–neuroimmune interplay as a potential player that drives itch in AD.

Here, we highlight the role of the neuroimmune response to obnoxious airborne toxins such as PM and its role in initiating severe pruritus in AD. In addition, we focus on the Fc Epsilon-driven proinflammatory response leading to itch. Finally, we suggest a couple of potential molecular immune mediators involved in the FC-Epsilon RI–neuroimmune pathway that could be targeted as treatment options to relieve itch in AD.

## 2. The Neuroimmune System and Skin

The bidirectional association between the neuroimmune system and skin diseases has been long recognized [10]. Neurotransmitters (e.g., Ach, NGF) or neuromediators (peptides) released by autonomic or sensory nerve endings and even by skin cells (e.g., keratinocytes) communicate with cutaneous immune cells and skin cells thereby inducing inflammation, skin barrier dysfunction and itch. There are 25 known neuromediators released by skin structural components including keratinocytes, fibroblasts, melanocytes, and endothelial cells [11]. These neuromediators include calcitonin gene-related peptide (CGRP), bradykinins, neurohormones, such as adrenocorticotropic hormone (ACTH), and melanocytes-stimulating hormone (MSH), catecholamines, endorphins, and acetylcholine. The levels of neuromediators are highly dependent on the physiological, pathological, and emotional status [12]. The skin has receptors that accept these neuromediators as well as enzymes that degrade them (Figure 1). Once bound to the receptor, a neuromediator induces skin functions at the cellular level including the immunological response. For instance, substance P triggers keratinocytes by boosting the production of cytokines (Interleukin-8 (IL-8), tumor necrosis factor alpha (TNFα)). In addition, sensory nerves (C-fibers) transmit the effect of skin stimuli to the central nervous system thereby inducing scratching, but may also contribute to negative emotional changes like depression and anxiety.

The skin, bidirectionally, modulates neuronal as well as immune signaling pathways. Therefore, skin can be visualized as a window that links cutaneous, endocrine, immune, and neuroimmune functions upon its activation by endogenous or exogenous stimuli in AD.

## 3. Neuroimmune Crosstalk in Skin Allergy

The immune and nervous system communicate interactively and closely. During the allergic response, skin cells become involved by being a rich source of neuromediators and being a target themselves for neuropeptides or neurotrophins as well as neurotransmitters in lesioned skin areas. Neuropeptides and neurotransmitters mediate this continuous communication by mediating the interactions with the immune system [13]. In turn, immune cells, which host the central and peripheral nervous systems, respond to the neurological triggers by secreting inflammatory cytokines.

In humans, both the sympathetic nervous system (SNS) and the parasympathetic nervous system (PNS) are rich in axons that conduct informative messages from and into the central nervous system (CNS) via afferent sensory and efferent motor neurons, respectively. The sensory and motor neurons have their own neurotransmitters and cytokine receptors and work by releasing neuropeptides [14]. Reciprocally, the neurotransmitters and the neuropeptides released by the PNS regulate the function of multiple immune cells. This interactive communication between the nervous system and immune system presents a key to inflammatory processes including allergy, infections, and tissue injuries [15].

Pruritus in AD presents as a protective mechanism to eliminate external stimuli driven by the immune system [16]. The IgE high affinity receptors are widely expressed on sensory neurons and can easily conduct signals to the CNS. This epidermal neuroimmune network can be viewed as a gateway for chronic inflammation and its association with tissue healing or remodeling.

## 4. Association of the Neuroimmune System with Atopic Dermatitis

Persistent chronic itch is a defining symptom in AD. Pruritus in itself is a protective mechanism designed to expel toxic skin substances [17]. However, it is associated with detrimental impacts on patients in terms of life quality, sleep interference, and lost productivity. Pruritus is mainly stimulated by networked communication between the cutaneous immune system, nervous system, and epidermal keratinocytes [6] and can be alleviated by immunosuppressants, or targeted antibody therapy (e.g., anti IL-4/13, anti-IL13, anti-IL31) [6]. Despite symptomatic improvement with glucocorticosteroids and immunosuppressants, these medications are not always effective when targeting the neurological part of itching, and sometimes necessitate combination therapies with neuromodulatory agents such as gabapentinoids, for example [6,18]. To better understand the neuro-immunological components involved in pruritus, we will first summarize the signaling pathways that reflect this link.

## 5. Neuronal Signaling Associated with Pruritus

Pruritogens (i.e., PM2.5 and PM10) can activate the skin’s sensory nerve endings by so far poorly understood mechanisms. The pruritogens attach to the afferent somatic neurons via pruritoceptors that innervate the skin and synapse to the projection neurons, modulated by interneurons in the spinal cord [19]. Activation of prurireceptors is followed by the influx of e.g., calcium ions and also activation of intracellular signaling pathways resulting in electrical impulses conducted from the skin to the spinal cord, as well as peripheral neuronal sensitization. The impulses are then received by the brain via the spinothalamic tract neurons where they are processed and an itch signal along with motor activity (scratching) is initiated.

The crosstalk between peripheral nerves (histamine-independent C-fibers), adaptive and innate immune cells, and e.g., keratinocytes or fibroblasts mediate pruritus in AD. Non-histaminergic pruritogens have different receptors and cutaneous nerve cells compared to histaminergic ones and majorly contribute to chronic scratching in AD. As a major environmental toxin, small PM can trigger non-histaminergic receptors [20] that require the calcium ion channels (TRPV1) and (TRPA1) to signal for pruritus to the spinal cord, which, in turn, coveys the message via gastrin-releasing peptide receptor (GRPR) neurons and brain-derived natriuretic peptide (BNP). The itch signals are then inhibited, in normal persons, by gamma-aminobutyric acid (GABA)-ergic interneurons. In AD patients, however, the GABA-ergic interneurons are probably either lost or downregulated [21].

### 5.1. The Role of Immune Cells

The role of immune cells in the itch cycle is best exemplified when Th2 cells, mast cells, and e.g., eosinophils amplify the pruritoceptive and inflammatory AD pathways by secreting neuro-peptides, chemokines and cytokines. The latter include thymic stromal lymphopoietin (TSLP) or IL-31 that initiate pruritus by activating TRPA1 and TRPV1 neurons that express their receptors [22]. Other cytokines are IL-4 or IL-13 (Campion M Exp Dermatol 2018; Oetjen Cell 2017), which work on sensitizing sensory neurons to external pruritogens including PM, among others. The AD flares have been documented to be associated with cytokine-to-neuron signaling by TSLP, IL-4, IL-13 and IL-31 [6,23].

### 5.2. The Contribution of Keratinocytes

Keratinocytes present a key pillar in pruritus development in AD. They release pruritogens such as thymic stromal lymphopoietin (TSLP) when they become triggered by mast cells that stimulate pruritoceptive neurons. Once TSLP is attached to the TSLP receptor expressed on type 2 innate lymphoid cells (ILC2) and type 2 immune cells (Th2 cells), the secretion of pruritogenic type 2 cytokines is initiated [22]. IL-4, IL-33, and IL-13 as well as neuropeptides (BNP, ET-1) increase the expression of TSLP on keratinocytes, opening the door for additional pathogenic pathways to be associated with pruritus. Keratinocytes were also found to intensively express pathogen-associated molecular patterns (PAMPs) from lesioned skin in AD patients [24].

### 5.3. The Neuronal Contribution

Cutaneous nerve endings release neuropeptides [22] such as BNP, CGRP or substance P (SP) thereby inducing neuroinflammation leading to modulation of immune cells thereby producing more cytokines, modulating skin barrier function by downregulating proteins involved in skin barrier protection, by inducing vasodilation and plasma extravasation (resulting in erythema and edema) as well as leukocyte recruitment aggravating inflammation [6,25]. This feed-forward mechanism probably also contributes to epidermal thickening and maybe fibrotic changes, as seen in prurigo lesions. In addition, substance P and its receptors, neurokinin-1, TRPV2, TRPA1, PAR2 and PAR4 are overexpressed in pruritic skin lesions [4].

## 6. Particulate Matter and AD Pruritus: Potential Neuroimmune Mechanisms

In order to better understand the molecular signaling pathways that link AD pruritus with the neuroimmune system in humans, this review first defines the impact of the long-term exposure to PM on AD pathogenesis.

### 6.1. The Role of PM in AD Pathogenesis

In this review, we focus on the noxious environmental air pollutants, specifically PM, that are associated with advanced industrialization and urbanization processes in developing countries in Asia and Latin America [26].

Increased levels of PM2.5 have been demonstrated to correlate positively with AD symptoms. In a report by Guo et al., 2018, a significant increment in outpatient visits for eczema and dermatitis was seen from April 2012 to 2014 per each interquartile rise in PM2.5 and PM10, along with other pollutants in women and elderly patients. In addition, [27] also demonstrated that the prevalence of eczema was associated with increased levels of PM2.5 and PM10.

Similar to other ambient air pollutants, PM2.5 dysregulates the skin barrier by generating reactive oxygen species (ROS) and polarizing Th2 immune cells [28]. This paves the way towards proinflammatory cutaneous immune initiation. PM has been also documented to disrupt the skin barrier via reducing the expression of epidermal structural proteins (E-cadherin, filaggrin, cytokeratin). Another potential mechanism of PM2.5 is through the activation of nuclear factor kappa B (NFkB) and the aryl hydrocarbon receptor (AhR) signaling pathway [29]. This signaling drives granulocyte infiltration and inflammation as well as pruritus via the production of the neutrophilic factor artemin.

### 6.2. Particulate Matter: Definition and Sources

Ambient PM is a common air pollution proxy indicator. PM has become a major hot subject of global concern with more attention gained in the mid-1980s. The PM refers to a complex mixture of liquid and solid particles that are air-suspended [20]. The major constituents of PM include mineral dust, SO_4_, NO_3_, NH_4_, NaCl, black carbon, and water. Ambient PM levels are affected by locally produced gaseous pollutants, geography, seasonal patterns, and meteorology.

PM can be grouped by size. Course PM comes from mechanical shearing, resuspended dust, and bioaerosols, such as spores and pollen, while fine PM is sourced from biomass burning, fuel combustion, and from secondary particles that result from the transformation of gaseous precursors in the atmosphere [26]. Other sources of fine PM include forest fires, vehicle exhausts, petrochemical industries, and power plants.

### 6.3. Particulate Matter Triggers AD Pruritus

To better conceive the connection between PM and AD itch, it is vital to observe the steady rise in atopic diseases in the developing countries where the worst air quality levels are recorded. According to the report of the third phase of the International Study of Asthma and Allergies in Childhood (ISAAC), the prevalence of AD in Latin America was 6–10% and in Africa 12–14%. The prevalence hit 7–27% among two-year-children in Asian Pacific countries [30] reflecting the impact of continuous skin exposure to noxious environmental irritants.

The skin comprises the first line of defense against extrinsic noxious environmental components including PM. Beyond this skin barrier, a complete network of somatosensory nervous system neurons converts sensory stimuli such as PM into itch sensation (pruriception). Sensory signals found in nerve endings bind to specific receptors that trigger membrane depolarization via transient receptor potential (TRP) channels [31]. When the signal is amplified, e.g., voltage-gated sodium (NaV) channels open, triggering an action potential [19]. The head, neck, and dorsal root ganglia (DRG) are rich in somatosensory neurons that receive signals from other organs. These neurons have one axonal branch that extends through the skin while the other branch projects towards neurons in the brainstem and spinal cord where information is conveyed to the cortex and the sensation is finally perceived.

The skin is innervated by various classes of sensory nerve endings. The cutaneous sensory nerves are stemmed from the cell bodies located in the dorsal root ganglia [32]. These nerve fibers are classified into Aβ, Aδ, and C-fibers based on their diameter, myelination and thus speed of signal transmission that conveys the itch sensation [33]. C-unmyelinated nerve fibers are the prevalentfibers associated with skin itch. With this significant overlap between neurosensation and itch, it was widely believed for years that a subtype of pain sensation mediated pruritus in atopic skin diseases. However, since the discovery of the itch-specific GRPR pathway in 2007, which functions independently of pain, the pruritus mechanisms have been proven to work separately [16].

### 6.4. Particulate Matter—FcεRI–Neuroimmune Interplay in AD Pruritus

Mast cells, found predominantly in the dermal layer of the skin, originate from hematopoietic progenitors in the bone marrow [34]. These cells lead a key role in the inflammatory and allergic responses via releasing histamine after being activated by the IgE-allergen immune pathway by air pollutants. The role of FcεRI in initiating pruritus is via linking IgE and responsive skin mast cells [35]. Allergic individuals produce anti-allergen IgE antibodies that attach to the high affinity FcεRI expressed on mast cells to initiate the activation of specific protein kinases (i.e., Fyn, Lyn, and SYK) (Figure 2). This step is followed by the phosphorylation of signaling proteins (LAT, PLC-γ, SLP-76). Phosphorylated LAT acts as a scaffolding protein and recruits other adaptor molecules including SRC homology2 (SH2)-domain-containing transforming protein C (SHC), GRB2-related adaptor protein (GADS), growth-factor-receptor-bound protein 2 (GRB2), and SH2-domain-containing leukocyte protein (SLP76), which signals the enzyme phospholipase C gamma 1 (PLC-γ1) and adaptor molecules (VAV, SOS) [36,37]. Once these enzymes and adaptors are activated, GTPases (RAC, RAF, RAS) become stimulated leading to the secretion of histamine and other pro-inflammatory cytokines (i.e., leukotrienes, prostaglandins) by mast cells. When FcεRI is stimulated, it initiates the PLCγ phosphorylation that, in turn, hydrolyses PIP2 to form IP3 and 1,2 diacylglycerol (DAG), which aids in releasing stored Ca^+2^ and stimulates PKC, respectively, leading, in turn, to histamine release by mast cells [38].

Clinically, the neuroimmune–FcεRIα modulation of itch has been recently studied in vivo and in vitro (Table 1). Many allergic disorders have been shown to have an FcεRIα involvement in their pathogenesis.

These ailments range from allergic rhinitis to food allergy and rheumatoid arthritis, among others, where neurons (i.e., trigeminal, enteric, etc.) work as targets for neuroimmune mediators.

## 7. Potential Treatment Lines That Target the Neuroimmune-Mediated FcεRI Neuronal Signaling Pathway in AD Pruritus

Due to the multiple molecular pathways involved in pruritus development, pharmaceutical treatment options have always been a challenge for clinicians. Most classical therapeutics target the inflammatory-driven itch using antihistamines combined with corticosteroids. To overcome this challenge, we propose, here, a collection of potential treatment lines that target the neuroimmune mediators specifically involved in the FcεRI signaling pathway. Table 2 presents some of these targets and their mechanisms of action.

## 8. Conclusions and Future Outlook

Here, we have shed light on the mounting body of evidence that demonstrates the pivotal role of the neuroimmune axis in regulating many atopic/allergic disorders including skin diseases such as AD.. As a link that connects the nervous system with the immune system within close circuits that connect skin cells, immune cells and nerves, pruritus is a hallmark of clinical AD, although pain is not as rare and may even occur with itch in patients. Relatively new is the evidence that pruritus in AD can be triggered by different allergens including environmental noxious air pollutants such as PM. The fine nature of PM makes it easy to penetrate disrupted human skin barrier, resulting in inflammatory responses including ROS generation, thereby initiating a strong neuroimmune response between the sensory nervous and immune system as well as epidermis thereby aggravating itch and scratching, inflammation and more skin barrier disruption.

Many neuroimmune mechanisms have been suggested to be involved in the cellular mechanisms related to pruritus; some of them still need to be validated in human diseases. Which cytokines are critical in which AD patient, and whether we can stratify each patient to an optimal personalized therapy, still needs to be clarified for AD or prurigo. Research has recently indicated that the FcεRI signaling pathway is highly associated with itch in atopic skin diseases. Here, we focused on the potential role of the FcεRI signaling pathway as a player in mediating pruritus related to long-term PM exposure in AD patients. The exact role and relevance of PM in the neuroimuune communication network of AD still needs to be exploed. Recent evidence suggests that mediators involved in the FcεRI signaling pathways may be used as potential pharmaceutical therapeutic targets for AD-associated pruritus in the future, by modulating the crosstalk between neuroimmune, immune and skin cells.

Pruritus in AD patients is induced by various environmental triggers such as irritants, allergens, proteases, bacteria, and most likely also airborne PM via the interplay between the immune and neural systems, as well as skin cells. Climate change is a significant thread globally. Thus, future studies may need to focus on the mecchanisms how climate change factors such as pollution impacts skin diseases including AD. For example, Fc-Epsilon RI–neuroimmune signaling pathway, with all of the involved inflammatory and immune players could be therapeutic targets for AD and pruritus. Investigations looking at the role of pollution on AD are currently under way.

## 9. Limitations

In this review paper, we used the most relevant research papers and reviews to support our proposal for the role of the Fc Epsilon signaling pathway, as a new perspective, for looking into the neuroimmune and inflammatory interplay in AD itch. However, we recommend that systematic reviews and metanalyses be conducted for future studies.

## Figures and Tables

**Figure 1 ijms-24-11851-f001:**
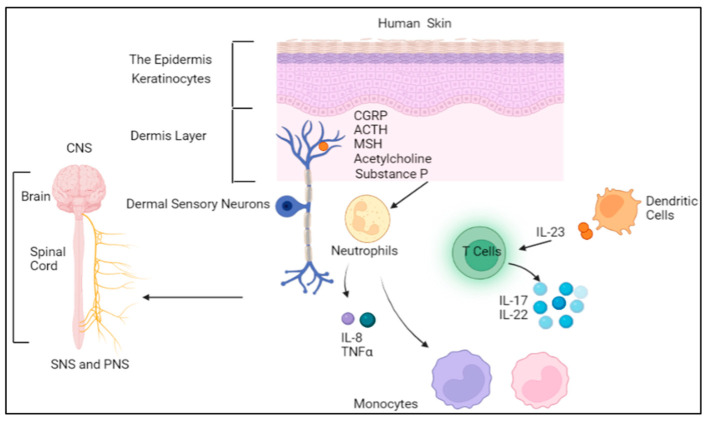
The neuroimmune system and skin. Keratinocytes produce neuropeptides (MSH, CGRP, Substance P) that trigger immune cells to produce proinflammatory cytokines. Abbreviations: SNS: sympathetic nervous system; PNS: parasympathetic nervous system; CNS: central nervous system; CGRP: calcitonin gene-related peptide; ACTH: corticotropin; MSH: melanocyte stimulating hormone; IL: interleukin; TNFα: tumor necrosis factor alpha.

**Figure 2 ijms-24-11851-f002:**
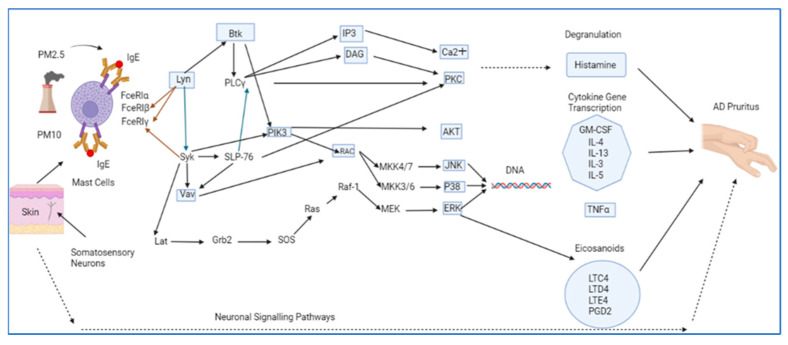
Potential Neuroimmune Signaling Pathway in AD Itch. The PM2.5 and PM10 trigger mast cells that express FcERI to bind to IgE. This is followed by activating signaling mediators that end up with histamine degranulation, proinflammatory cytokine production, and eicosanoids release leading to AD pruritus. Abbreviations: PM: particulate matter; IgE: immunoglobulin E; FceRI: high affinity IgE receptor; Syk: spleen tyrosine kinase; LAT: large amino acid transporter 1; Grb2: growth factor receptor-bound protein 2; Ras: rat sarcoma virus protein; Slp-76: Src homology 2 (SH2) domain-containing leukocyte protein; Btk: Bruton’s tyrosine kinase; IP3: inositol triphosphate; DAG: diacylglycerol; Ca++: calcium ions; PKC: protein kinase C; AKT: protein kinase B; Raf-1: rapidly accelerated fibrosarcoma; MKK: mitogen-activated protein kinase kinase; IL: interleukin; TNFα: tumor necrosis factor alpha; LTC: leukotriene C; LTD: leukotriene D; PGD2: prostaglandin D2; JNK: Jun-N terminal kinase; P38 MAPK: mitogen-activated protein kinase; ERK: extracellular signal-regulated kinase; MEK: mitogen-activated protein kinase kinase; DNA: deoxyribonucleic acid.

**Table 1 ijms-24-11851-t001:** Reports on the Association between Allergy Conditions and the FcεRIα Neuronal Signaling Pathway.

Allergic Condition	Study Model	Target Neurons	Target Mediator(s)	Results	Reference
Allergic Conjunctivitis	mice	Trigeminal neurons	IgE–immune complex, FcεRIα	FcεRIα was expressed in conjunctival sensory neurons; The IgE immune-activated trigeminal neurons. Both actions were recorded in the absence of any inflammation	[39]
PM_2.5_ sensitization	Bone marrow-derived mast cells (BMMC)		IL-6, IL-13, MCP1, TNFα, Syk, LAT, SLP76, PLCγ1, FcεRIα	BMMC exposed to low levels of PM_2.5_ showed mast cell degranulation and FcεRI-modulated cytokine production	[35]
Rheumatoid Arthritis (RA)	Sprague–Dawley rats		FcεRIα	FcεRIα expression was increased in dorsal root ganglion neurons (IB4 and TRPV1) in RA rats and was reduced in knockout rats	[40]
Food Allergy (FA)	BALB/c mice	Enteric neurons	FcεRIα	FcεRIα was highly expressed in enteric neurons where mast cells were allocated in FA mice	[41]
Allergic Rhinitis (AR)	AR patients	GINIP neurons	TAFA4, IgE, FCER1G	The transcription of Fcer1g was restricted by TAFA4 in mast cells via the TAFA4–PTEN-PU.1 axis in airway tissues, and suppressed AR symptoms	[42]
IgE sensitization	Bone marrow-derived mast cells (BMMC)	Dorsal root ganglion	IgE, TNFα. IL-6, CADM1	The sensory neuron–BMMC interaction induced mast cell degranulation and enhanced the responses to antigen stimulation and the activation of FcεRI receptors	[43]

**Table 2 ijms-24-11851-t002:** Reports on Selected Inhibitors that Suppress Target Mediators in FcεRI Pathway.

Target Mediator(s)	Inhibitor(s)	The Inhibitor’s Mechanism of Action	Diseases Treated by the Inhibitor	Selected References
IgE–FcεRI	Omalizumab	Omalizumab suppress IgE–FcεRI attachment resulting in less free FcεRI available to receptors on antigen presenting cells, mast cells, and basophils	Allergic asthma, chronic spontaneous urticaria (CSU), and nasal polyps	[44]
Lyn	Dasatinib	Fostamatinib inhibits the binding of CD19 with Lyn and PI3K p85	Myeloid leukemia	[45,46]
Syk	Fostamatinib	Selectively targets the Syk signaling pathway by inhibiting the signal transduction of Fc-activating receptors and B-cell receptors.	Rheumatoid arthritis and immune thrombocytopenia, B-cell lymphomas	[47,48]
LAT1	BCH	BCH Inhibits neutral amino acid transfer (leucine) by LAT1	cholangiocarcinoma	[49,50]
Btk	Acalabrutinib	Acalabrutinib binds to cysteine 481 in the ATP-binding site of BTK, located in the kinase domain	chronic lymphocytic leukemia	[51]
SLP-76	PP1, PP2	PP1 and PP2 bind to an area of SLP-76 that does not overlap with the ATP-binding domain	PTK6-positive malignant diseases	[52]
Ras	Sotorasib	Sotorasib attaches to inactive KRAS covalently between cysteine 12 and acrylamide, and non-covalently between the isopropylpyridine substituent and Histidine H95, Tyrosine Y96 and glutamine Q99	NSCLC with K-RAS (G12C) mutations	[53,54]
PDK1	GSK-470	GSK-470 suppresses the Akt1 activation by PDK1 in the presence of fat vesicles that contain PtdIns [3,4,5] P3 or an Akt 1 mutant	Multiple myeloma	[55]
RAC	MLS000532223 EHT1864	Stop the activated Rac 1 and Rac-independent Tiam 1-cell transformation as well as Ras transforming proteins	Cancers	[56,57]
Raf-1	Vemurafenib and Dabrafenib	Vemurafenib and Dabrafenib work on the ATP competitive binding of the active conformation of BRAF kinase	metastatic melanoma	[58]
MEK	Trametinib	Trametinib blocks MEK1/2 kinase activity and prevents RAF-dependent MEK phosphorylation	non-small cell lung cancer, metastatic melanoma	[59,60]
PKC	Sotrastaurin	Sotrastaurin inhibits the PKC α, β and the θ isoforms resulting in selective NF-κB inactivation	autoimmune disease and transplant organ rejection	[61]
BTK	Ibrutinib	Ibrutinib blocks and inactivates BTK though a covalent bond that connects ibrutinib with cysteine 481 in the ATP-binding site irreversibly	chronic lymphocyte Leukemia	[62,63]
ERK1/2	KO-947	KO-947 selectively inhibits ERK1/2 in the RAS/MAPK signaling pathway	Solid tumors	[64]
Fyn Kinase	Saracatinib	Saracatinib blocks the ATP-binding active site of the kinase domain bond	Alzheimer’s disease, Cancers	[65,66]
PI3K	Alpelisib	Alpelisib inhibits PIK3 selectively in the PI3K/AKT kinase signaling pathway	Breast cancer, PIK3CA-related overgrowth spectrum	[67,68]
cPLA2	Arachidonyl trifluoromethyl ketone (ATK)	AKT is an arachidonic acid analog where the carboxyl group is replaced by a trifluoromethyl ketone (TFMK) group; therefore, suppressing cPLA2	Cancers	[69]
AKT	Capivasertib	Capivasertib is a selective adenosine triphosphate (ATP)-competitive inhibitor of AKT	Breast cancer	[70,71]

Abbreviations: IgE Fc Epsilon: immunoglobulin E Fc Epsilon; Syk: spleen tyrosine kinase; LAT1: large amino acid transporter 1; BtK: Bruton’s tyrosine kinase; Slp-76: Src homology 2 (SH2) domain-containing leukocyte protein; Ras: rat sarcoma virus protein; Raf: rapidly accelerated fibrosarcoma; MEK: mitogen-activated protein kinase kinase; PKC: protein kinase C; PTK: protein tyrosine kinase; ERK1/2: extracellular signal-regulated kinase; Fyn Kinase: SRC family tyrosine kinase; PI3K: phosphatidylinositol-3 kinase; cPLA2: cystolic phospholipase-2; AKT (aka PKB): protein kinase B.

## Data Availability

Correspondence and requests for materials should be addressed to Sergio Crovella.

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
