# Peer review of "Fc Epsilon RI–Neuroimmune Interplay in Pruritus Triggered by Particulate Matter in Atopic Dermatitis Patients"

_ijms, 2023, doi:10.3390/ijms241411851_

Round 1

Reviewer 1 Report

This review enlights the pathways of pruritus in atopic dermatitis (AD), with fluent English and clear explanations of the different molecules and receptors involved in itching. Additionally, the authors provide insight into the role of pruritogenic mediators (PMs) in the genesis of pruritus in patients with AD.

However, there are some issues with the citation format. Citations should be sorted in the order of their appearance in the text, rather than in alphabetical order. In the article itself (not in the bibliography), citations in square brackets should follow numbers in ascending order, such as [1], then [2], [3], and so on. They should not be presented in a different order.

Another concern is the inconsistent use of square brackets. If the authors use square brackets for citations, they should avoid using them for other purposes such as acronyms or abbreviations, as this can create confusion for readers.

Author Response

Dear Sir,

Thank you for the valuable remarks. I did the following:

  1. I kept all references in the text between this tyle of brackets().
  2. I inserted the references based on the their ascending order.
  3. I used this tyle for brackets [] was used for other abbreviations and explanations. 
  4. I left the bibliography list empty as I need to add more references to the manuscript. 

Reviewer 2 Report

The paper is interesting and presents an intriguing outline on the state-of-the art of a very relevant topic. However, I would suggest the authors to better describe the methodology related with this research, in particular stressing out the search strategy, inclusion/exclusion criteria, and eventually following the PRISMA guidelines (eventually releasing a dedicated plot) to be sure that the methodology followed is robust and reliable.

Then, a stronger take-home message to the reader is needed, together with clear indications about future developments expected in the field, stemming from the review retrievals.

Furthermore, there are some missing references throughout the manuscript, leading to important typos. Please, double-check and correct.

Author Response

Dear Sir,

Thank you for the comments. I did the following corrections:

  1. added rationale and aims under the production part.
  2. added a take-home note and future outlooks in the conclusion.
  3. I did not include the references as I need to add more ones within the text and share it with all reviewers.
  4. I have a small question: for (methodology related with this research, in particular stressing out the search strategy, inclusion/exclusion criteria, and eventually following the PRISMA guidelines (eventually releasing a dedicated plot) to be sure that the methodology followed is robust and reliable), how can I apply this to my manuscript since it is a pure review and not a research paper nor a systematic review/ meta analysis. Can you kindly elaborate on this point? 

Reviewer 3 Report

In the present era, people suffering from allergic symptoms are increasing. One of the reasons why that is some kinds of artificial chemicals and materials. In this review, the authors shed light on PM particles to provoke allergic reactions via the FCεRI receptor in view of neuroimmune interplay. The review article will be meaningful for IJMS readers. However, some questions should be responded to by the authors, below.

The authors referred to the exposome and envirome as “this review first defines the impact of exposome and envirome on AD”. Considering the position of this chapter, the reviewer wonders if this is not “first” section. In addition, exposome is defined as “the sum of external factors”, but the central interest in this review is specifically PM. This explanation is the odd one out in this entire text. The reviewer thought this introduction of exposome and envirome is not mandatory for the latter descriptions of PM.

Minor points,

l  Section 2, Line 91:

the production of cytokines [IL-8, tumor necrosis factor [TNF-α] --> the production of cytokines [IL-8, tumor necrosis factor alpha [TNF-α]]

l  Section 6.4, Line 250;

Ca+2 --> Ca2+

Author Response

Dear Sir,

Thank you for the valuable remarks!. I did the following corrections:

  1. removed the sentences related to envirome and exposome as they may look irrelevant.
  2. I corrected the TNF and Ca+2 .
  3. May I kindly ask you to guide me improve my scores on points 2 and 2 as you rated me with few yellow stars only ? Thank you sir again.  

Round 2

Reviewer 1 Report

This review enlights the pathways of pruritus in atopic dermatitis (AD), with fluent English and clear explanations of the different molecules and receptors involved in itching. Additionally, the authors provide insight into the role of pruritogenic mediators (PMs) in the genesis of pruritus in patients with AD.

Finally, citation format is appropriate and clear now. Even conclusions have also been improved. 

Author Response

Thank you sir for the remark. Bless your heart. 

Reviewer 2 Report

The paper needs revision in terms of the references (as the authors stated in their response), but its quality is not bad.

As for the clarification about PRISMA, I am aware of the authors' concerns. Of course, this diminishes the "excellence" of the product, therefore my suggestion (unless the authors want to change their format) is to keep such considerations under the limitations section with a brief related discussion.

Author Response

Dear Sir,

thank you very much for sharing your valuable remarks.

1.I have added 23 more references to the manuscript and ordered them as per their appearance in the text. 

2.How may I improve the 2nd and 3rd points where I have received 3 stars only?

3. At the end of the manuscript I added a small paragraph on the method I used and the limitation that future papers may focus on (i.e. meta-analysis and systematic review). 

Reviewer 3 Report

2. The minor points were all corrected.

1,3. The authors considered the expression to more organized manuscript. The manuscript owes clarity of the assertion to the reconsideration and it helps potential readers to realize the matters. Hence, stars improved.

Author Response

Dear Sir,

Thank you very much for the remarks. Bless your heart!

Round 3

Reviewer 2 Report

The authors have paid a lot of efforts to improve their paper and they addressed all of my concerns, overall. They frankly acknowledged their study limitations concerning the methodology, providing a tip for future works in the field. I recommend the acceptance of the present work, once problems with citations in the text are solved with the support of the Editorial Office.